# Is There a Promising Market for the A2 Milk? Analysis of Italian Consumer Preferences †

**Deborah Bentivoglio**, **Adele Finco** *, **Giorgia Bucci** and **Giacomo Staffolani**

Department of Agricultural, Food and Environmental Sciences (D3A), Università Politecnica delle Marche (UNIVPM), Via Brecce Bianche, 60131 Ancona, Italy; d.bentivoglio@staff.univpm.it (D.B.); g.bucci@pm.univpm.it (G.B.); gstaffolani95@gmail.com (G.S.)

* Correspondence: a.finco@univpm.it; Tel.: +39-071-220-4930

† These investigations and many of the results will be presented at the XVI EAAE Congress in Prague, 20–23 July 2021.

**Abstract:** Milk consumption has been on the decline for decades. Part of the cause for this is due to changes in consumer behavior and interest in healthy eating. The aim of this study was to estimate the premium price that consumers are willing to pay for A2 milk. A2 milk is a newcomer type of product containing only A2 beta-casein that is spreading in many countries of the world as a functional food. In addition, it is produced by traditional breeds of cattle that are adapted to the marginal areas and could be a virtuous model of the sustainable management system to which the consumer lately seems to turn in an increasingly conscious way. To accomplish this aim, contingent valuation has been used, which permits using a survey and a direct estimation of the premium price that consumers are willing to pay (WTP). The question format used is a dichotomous choice valuation based on a double-bound model. Statistical analysis was carried out using descriptive statistical analysis. The empirical results reveal that consumers are willing to pay a premium for A2 milk. In addition, individuals interested in product quality and already buying functional products are attracted to this type of product.

**Keywords:** A2 milk; milk sector; functional food; contingent valuation; WTP; consumer; marketing strategy; sustainable development

## 1. Introduction

The European Union (EU) dairy sector is the second biggest agricultural sector in the EU, representing more than 12% of total agricultural output. Over the past two decades, milk production in the EU has been steadily increasing. It increased from 152 million tons in 2000 to over 172 million tons in 2018 (an increase of almost 11%). The most likely explanation for this is that the quota system introduced in 1984 with the objective to bring increasing milk production under control was abandoned at the end of March 2015. All 28 Member States produce milk. The main producers of cow milk are Germany (19%), France (15%), the United Kingdom (9%), the Netherlands (8%), Poland (8%), Italy (8%), Spain (5%), and Ireland (5%), which together account for three-quarters of total EU production. The remaining Member States produce just a quarter of EU production. Most of the milk produced is delivered to dairies for further processing (93%). Among the products derived from raw milk, drinking milk is the only one that registered a regression in production. Among the products derived from raw milk, drinking milk is the only one that registered a regression in production. Drinking milk includes raw milk, whole milk, semi-skimmed, and skimmed milk containing no additives. Relates only to milk directly intended for consumption, normally in containers of 2 L or less. It also includes milk with vitamin additives. In detail, the production has gone from 31 million to 29 million tons from 2013 to 2018, with an average annual loss of 0.3 million tons [1].

In line with the negative trend of production of drinking milk, the EU per capita consumption also recorded a decrease annually. In particular, the volume of milk consumed dropped by 2% between 2013 and 2018. According to EU agricultural outlook 2018–2030, European liquid milk consumption is expected to continue declining in the EU. Campaigns promoting lower dairy product intake because of the climate and environmental footprint of livestock products, as well as an increase in lactose intolerance claims, will have a downwards influence on the consumption of dairy products. Consumer demand, in fact, drastically changed in recent years: consumers take more interest in aspects such as nutrition, health, and quality of the foods. Consequently, with declines in per capita milk consumption and changes in consumer preferences, the dairy industry has to be creative and innovative in developing products to increase milk sales [2,3]. Nowadays, functional foods are one of the most dynamic and innovative categories in the food industry, with an estimated global value of over 40 billion US dollars and steady annual increases in sales [4]. The markets for this new type of food are more developed in the USA and Japan than in the EU, in which Germany, France, United Kingdom, and the Netherlands represent the most important countries within the functional foods market; meanwhile, in Italy, the overall category showed a negative performance, not only in 2019. To date, functional foods have not yet reached a precise definition within European law. Thus, we may consider the definition coined by the International Food Information Council (IFIC). Functional foods are "foods or food components that may provide benefits beyond basic nutrition". More specifically, the Functional Food Center defines these food products as "Natural or processed foods that contain biologically-active compounds; which, in defined, effective, non-toxic amounts, provide a clinically proven and documented health benefit utilizing specific biomarkers, for the prevention, management, or treatment of chronic disease or its symptoms".

Thus, in the U.K., Australia, New Zealand, and recently, California, A2 milk has been introduced by the A2 Milk Company with a specific logo. The A2 Milk Company commercializes intellectual property relating to A1 protein-free milk that is sold under the A2 and A2 Milk brands, as well as milk and related products like infant formula. This type of milk is sold as a functional dairy food due to natural health benefits. A2 milk represents a relative newcomer to the ever-expanding health food market. A2 milk is a variety of cows' milk that mostly lacks a form of β-casein proteins called A1, and instead, has mostly the A2 form. Originally, all cows' milk was of the A2 type. However, a genetic mutation, probably happening between 5000 and 10,000 years ago, has resulted in a proportion of cows of European breeds producing a casein variant called A1 beta-casein. The proportion of A1 beta-casein is higher in the black and white breeds compared to the yellow and brown breeds, such as Pezzata Rossa and Bruna breeds. A1 beta-casein is absent in the milk of pure Asian and African cattle.

The interest in this type of milk is especially connected to the positive benefits to human health. From 1955 to date, different studies (Figure 1) have compared especially the effects of milk containing only the A2 β-casein type with milk containing A1 β-casein in humans.

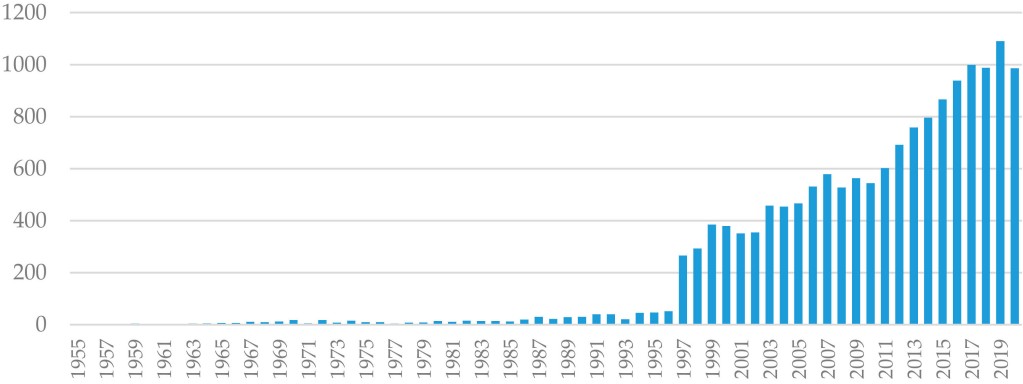

**Figure 1.** Documents by year from Scopus and Science Direct database, using the keywords "β-casein" AND "milk".

Some studies have reported health benefits from drinking A2 milk [5–11]; others show no evidence [12,13]. In addition, the European Food Safety Authority (EFSA) reviewed the scientific literature and published their results in 2009, finding that it could not be established that bioactive peptides in milk containing both the A1 and A2 proteins have an adverse effect on health.

Thus, to date, the debate on the benefit of A2 milk consumption is controversial. Nevertheless, farmers in many countries of the world are being incentivized to produce A2 milk to meet the growing demand in what is considered to be a healthier alternative to conventional dairy [14]. Only by stimulating farmers to produce differentiated milk can this counter the phenomenon of homologation of all dairy products, guaranteeing greater income also for small-sized farms located in the marginal area. Resilience and sustainability are keywords for the future of the dairy sector. This can be achieved with innovation as a way to reconcile the need for farmers to earn a decent living, consumer demand for healthy and quality dairy products, and environmental/animal health requirements [15,16]. The original A2 milk is produced by breeds that are typically the more "traditional", such as Pezzata Rossa and Bruna cows, that can be raised in marginal areas and which have not been as greatly impacted by over-breeding. However, the incessant process of farmland abandonment in mountain areas, due to the limited degree of mechanization and intensification, lack of structures and services, put at risk these productions. Hinojosa et al. [17] named this phenomenon "the mountain effect", which also implies the lack of High Nature Value farming [18] and the reduction in pastures, thus, generating a loss in biodiversity and Ecosystem Services (ES) provided by mountain areas [19]. In this sense, the potential and possibilities of producing this type of milk could ensure sustainable farming practices, animal welfare, the support of local economies, as well as the conservation of genetic diversity in native breeds. However, the decisions are complex because of uncertainties about future premiums for A2 milk [20].

In recent years, this type of milk has also appeared in Italy without a specific logo and brand. Understanding the demand for this differentiated product constitutes a cornerstone for further analysis of consumer choices and price competition. Related to consumers, the case of Brazilian consumers provided by Oliveira Mendes et al. [21] constitutes one of the few studies on the perception of A2 milk. According to the study, consumers recognize the difference between a cheese with A1 and A2 beta-casein, preferring the latter. In this context, the present paper, intends to answer the following research questions: which factors influence milk consumer buying behavior? Do Italian consumers know functional foods and consume them? Are consumers willing to pay a premium price for A2 milk? Are they willing to recognize an added value? This study was developed in the framework of the project I-MILKA2—*Innovation in dairy production using A2A2 Milk and assessment of the beneficial effects on human health*. The aim of the project is to protect the genetic biodiversity of dairy cattle with A2A2 genotype and measure the impact of milk production with beta-type A2 on consumer health, on cheese processing efficiency, and on quality dairy products. The project includes the analysis of the qualitative and technological parameters of A2 milk and its derived products, the evaluation of possible beneficial effects on human health of beta-casein A2 through studies on human adipose cells in culture, and finally, the evaluation of consumer preferences for the purchase of A2 milk and derivatives. In order to answer research questions, a contingent valuation has been used, which permits using a survey and a direct estimation of the premium price that consumers are willing to pay. In addition, the model employed aimed to identify which consumer's characteristics and other variables have an influence on the values derived for WTP. Thereafter, the paper is developed as follows. In Section 2, the data and methodology are introduced. The results obtained are shown in Section 3 and discussed in Section 4. The paper concludes by summarizing the findings of the case study and by presenting an analysis of their relevance for any future marketing strategies of milk associated with specific characteristics.

## 2. Materials and Methods

### 2.1. Consumer Questionnaire

This research is based on a survey sent out to estimate consumer WTP for A2 milk. The data collection survey was made up of four sections, comprising 30 questions. The first section asked about sociodemographic characteristics (e.g., age, household income, gender, etc.). The second section comprised questions devoted to milk consumption (e.g., purchase frequency, place of purchase, type of milk) and motives influencing milk purchase. About purchasing factors, respondents were initially asked to indicate the three main attributes deemed significant when buying milk. Then, the question was repeated, asking the interviewees to assign to the above factors, on the basis of a Likert scale, a value of importance from 1 to 5 (where 1 is not at all important, 3 is indifferent, and 5 is very important). After this section, knowledge and consumption of functional foods were addressed in section three. In this segment, respondents were also asked questions related to the consumption of functional milk. The questionnaire finished with a set of questions designed to obtain the WTP for A2 milk with respect to fresh lactose-free milk. We chose this product because today, A2 milk is promoted and sold in supermarkets as an alternative for people who struggle to digest common varieties of cows' milk and in some countries, it has been introduced as a functional dairy food due to natural health benefits.

The questionnaire was distributed through social networks (Twitter, Facebook), instant messaging apps (Facebook Messenger and WhatsApp), and through mailing. The use of an online survey brings with it numerous advantages such as the speed of diffusion, the ease of creating the database with the answers, and the ease of processing; on the other hand, it makes it impossible to reach people who do not have this technology [22]. In addition to this risk, it is possible that the sample is not very representative of the reference population (consumer universe), but through a wide sample, we were able to collect the main characteristics of the population [23]. According to Fujita et al. [24], in order to ensure the statistical reliability of WTP estimations, at least 400 samples for a double-bound are needed. Finally, respondents were sent a unique link to the website on which they simply had to click to start the online survey. This unique link ensured that the same person could not participate more than once.

The eligibility criteria to complete the questionnaire are represented by the answers to the questions: "Do you consume milk?" and "Do you buy functional foods?". In the case of a negative response, the interview was not conducted.

Before the survey was launched online, the questionnaire was pre-tested. Pre-testing was conducted through a small focus group composed of 25 persons assembled to discuss its understanding of and reaction to the questionnaire prior to the study. Persons included in the pre-test usually consume both milk and functional foods. The purpose of the pre-test was to assess if there was a need to make modifications on the designed questionnaire [25]. Based on the pre-test, some questions of the survey were restructured and the willingness to pay section was polished so that it became easier to understand by the respondents.

### 2.2. Methodology

The Contingent Valuation Method (CVM) was originally developed in environmental economics [26], but it is well suited to soliciting consumers' WTP for a product that is not yet on the market [27] and it is often preferred over other methods, such as experimental markets, for its flexibility and limited costs [28]. In fact, there are many examples in the literature which use CVM to estimate the importance for consumers of indication of origin of the product, quality certifications, or a particular method of production (organic) indicated on the label [29–39]. The contingent valuation consists of creating a hypothetical market for a non-market good in which the consumer expresses the WTP for buying it, which corresponds to the utility derived from its consumption. In the case of choosing between two products, the WTP is the maximum amount of money an individual is willing to give up to exchange a product with quality q0 for a product with quality q1 [40] and it

corresponds to the difference between surplus consumers before and after the improvement of one or more attributes of the product. The decision process is modeled using a random utility approach, in which consumers make their choices by identifying alternatives that provide the highest level of utility with the assumption that they know their preferences with certainty, but they are unknown to those who carry out the survey [41]. The utility

$$U = f (y, x, m) \tag{1}$$

is assumed to be a function of the consumers' characteristics (y), product attributes (x), and the consumer income level (m). According to Lancaster's Demand Theory [42], consumers are hypothesized to derive utility not directly from goods but from a collection of characteristics or attributes those goods possess [43]. A consumer will agree to buy a product of quality q1 at the offered price if the utility with the proposed change is greater than the utility derived from choosing the product of quality q0. Therefore, U is decomposed into an observable part and an error term (not observable). Therefore:

$$U (yj, x, m) = V (yj, x, m) + ej \tag{2}$$

where ej is a random error component and it is assumed to be independently and identically distributed with a mean of zero.

The CV approach allows a direct estimation of WTP by means of different elicitation techniques. In this study, we apply a Dichotomous Choice Contingent Valuation Method (DC-CVM) based on a double-bound model (the first question is followed by another specifying a lower amount if the answer to the first question was negative, and higher otherwise). In the double bound model, the interval is enclosed within two binds. The level of the second bind is contingent upon the response to the first bind. If the individual responds "yes" to the first bind, the second bind is some amount greater than the first bind; if the individual responds "no" to the first bind, the second bind is some amount smaller than the first bind. Thus, there are four possible outcomes: (1) both answers "yes"; (2) both answers are "no"; (3) a "yes" followed by a "no"; and (4) a "no" followed by a "yes". This model is the most suitable for the case in which the individual does not have sufficient information to attribute the value that the product could have if the market really existed. Furthermore, this format is simpler for the person interviewed and more realistic, since it corresponds to the real market situation in which the consumer is offered a product at a given price and must decide whether to buy or not. In addition, according to Hanemann et al. [44] and Leòn [45], this model is asymptotically more efficient than the simple model.

Assuming a linear functional form for the WTP, the econometric model [46,47] used in this study is the following:

$$Y = x'\beta + \varepsilon \tag{3}$$

where Y is the true individual WTP, which is assumed to depend on certain individual characteristics, such as income, and on individual socioeconomic characteristics contained in the vector x′; the term $\varepsilon$ represents the stochastic error, with zero average and a certain variance v2. In this model, Y is considered a latent continuous censored variable and, based on the four possible answers of the interviewee, it can be included in one of the following intervals:

$$
\begin{aligned}
\Pr (\textit{yes, yes}) &= \Pr (E \geq t^a_i \geq t_i) = 1 - F (t^a_i) \\
\Pr (\textit{yes, no}) &= \Pr (t_i \leq E \leq t^a_i) = F (t^a_i) - F (t_i) \\
\Pr (\textit{no, yes}) &= \Pr (t^b_i \leq E \leq t_i) = F (t_i) - F (t^b_i) \\
\Pr (\textit{no, no}) &= \Pr (E \leq t^b_i \leq t_i) = F (t^b_i)
\end{aligned}
\tag{4}
$$

The log likelihood function for this model is:

$$LogL = \Sigma^n_{i=1} I_i I_i^a \log[F(t^a_i)] + I_i(1 - I_i^a)\log[F(t^a_i) - F(t_i)] + I_i^b(1 - I_i)\log[F(t_i) - F(t^b_i)] + \\ (1 - I_i)(1 - I_i^b)\log[F(t^b_i)] \tag{5}$$

where the symbol $t_i$ is the starting price (1.89 EUR); $t^a_i$ is the price offered in the follow-up in the case of the first affirmative answer (1.99 EUR); $t^b_i$ is the price offered in the follow-up in the case of the first negative answer (1.79 EUR). The dichotomous variables $I_i$, $I_i^a$, and $I_i^b$ assume a value of one or zero depending on whether the answers to the initial question and to the corresponding follow-up have been positive or negative.

It is important to underline that the problems of the CVM include the hypothetical and starting point for price bidding. In the literature, several models, based on different assumptions, have been proposed to deal with this problem [48–51]. The double-bounded dichotomous choice contingent valuation model and pre-testing can reduce the biases [52]. In this study, the initial bid is derived from a market analysis from similar products, id est different brands of fresh lactose-free milk. The market price was used as a starting point, since consumers use reference prices as a threshold to evaluate the selling price of products, which on its turn, impacts subsequent purchase decisions [53]. Successively, the design and the ranges of the bid value sets were determined and tested from the focus group discussion and the pre-test survey.

The explanatory variables considered in the estimation are:

1. Socioeconomic variables affecting the respondent:

   - Female: dummy variable that takes value 1 if the respondent is female and 0 otherwise;
   - Age: in 6 groups (1 = <20; 2 = 20–29; 3 = 30–39; 4 = 40–49; 5 = 50–59; 6 = >60);
   - Education: a variable that takes values from 1 to 6 and correspond from lower to higher education (1 = No formal education; 2 = Elementary school; 3 = Middle school; 4 = Highschool: 5 = University; 6 = Ph.D. or higher);
   - Employment status: a variable that takes values from 1 to 5 (1 = employed; 2 = household; 3 = student; 4 = retired; 5 = unemployed);
   - Marital status: Dummy variable that takes value 1 if the respondent is married or cohabiting and 0 otherwise;
   - Family unit: is the number of family members of the respondent;
   - Income range: the variable that takes value from 1 to 6 and corresponds to increasingly high-income brackets (1 = >10,000 EUR; 2 = 11,000–20,000 EUR; 3 = 21,000–35,000 EUR; 4 = 36,000–50,000 EUR; 5 = 51,000–75,000 EUR; 6 = >75,000 EUR);
   - Region of origin: a variable that takes values from 1 to 20 corresponding to Italian Regions;
   - Residence: dummy variable that takes value 1 if the respondent resides in the urban area and 0 otherwise.

2. Variables connecting to the respondent purchasing behavior of milk: frequency of purchasing, which may take values from 1 to 5 (more than three, three, two, one, less than once a week); place of purchasing, which may take values from 1 to 5 (supermarket/organized large scale distribution; retail; local market; local producers; raw milk automatic dispensers); different types of milk based on percentage of fat which may take values from 1 to 3 (whole milk; semi-skimmed milk; skimmed milk); different categories of milk, which may take values from 1 to 10 (raw milk; fresh milk; high quality milk; microfiltered milk; UHT milk; organic milk; mountain milk; lactose-free milk; special milk; flavored milk).

3. Variables that include factors/attributes influencing the purchase of milk: Organic production, Mountain Origin, High Quality; Expiration data; Nutritional content; Organoleptic characteristics; Origin; packaging; Price; Brand—they assume values from 1 to 5 (on a 5-point Likert scale).

4.　　Variables related to respondent purchasing behavior of functional foods: Knowledge of functional foods, consumption of functional foods in the future, purchase of functional milk (dummy variable that takes value 1 if the respondent chooses the first alternative or 0 otherwise).

Both the dependent variable and the explanatory variables "Family unit" and "Region of origin" are expressed in logarithms. The estimates are obtained through the software Gretl version 1.9.4.

## 3. Results

### 3.1. Descriptive Statistics

This section presents the results of the study. Of a total of 1277 Italian consumers interviewed between October and December 2019, 1140 consume milk and of these, 899 have declared consuming functional foods. The sample is representative of the population for age, sex, education, and income in Italy. Statistical analyses of the coded questionnaire data were performed using Gretl 1.9.4. The sample is distributed throughout the national territory, with particular concentration in the center of Italy. The sample profile and reference population figures are given in Table 1. The data indicate that consumers are mainly females (67%), with a good level of education (40% have a diploma, 35% have a degree, and 19% have a postgraduate degree), with people of varying ages, unmarried (53%), and if married, with medium families (three or four persons per family). They are also most frequently residents of urban areas (62%) and their stated income included mainly between 21,000 and 35,000 EUR (36%).

**Table 1.** Socio-economic characteristics.

| Variables | Description | % | Variables | Description | % |
|---|---|---|---|---|---|
| Gender | Male | 33% | | 1 | 8% |
| | Female | 67% | | 2 | 23% |
| Age | <20 | 3% | Family members | 3 | 26% |
| | 20–29 | 33% | | 4 | 31% |
| | 30–39 | 23% | | 5 | 9% |
| | 40–49 | 19% | | 6 | 3% |
| | 50–59 | 15% | | >7 | 1% |
| | >60 | 6% | | No formal education | 0% |
| Marital status | Married | 47% | Education level | Elementary school | 0% |
| | Single | 53% | | Middle school | 7% |
| Residence | Urban | 62% | | Highschool | 40% |
| | Peri Urban | 38% | | University | 35% |
| Household income | >10,000 EUR | 5% | | Ph.D or higher | 19% |
| | 11,000–20,000 EUR | 24% | Professional activity | Employed | 66% |
| | 21,000–35,000 EUR | 36% | | Housewife | 4% |
| | 36,000–50,000 EUR | 19% | | Student | 22% |
| | 51,000–75,000 EUR | 9% | | Retired | 2% |
| | >75,000 EUR | 7% | | Unemployed | 5% |

Before turning to the WTP estimation (section four), we take a closer look at milk purchasing behavior (second section) and also, the knowledge and the consumption of functional foods (third section). In total, 58% of the sample buy milk 1 or 2 times a week, 24% buy it 3 or more times a week, while 18% buy it less than once a week. Most of the milk (93%) is purchased in supermarkets/GDO. Based on the fat content, the most consumed type of milk is the partially skimmed, which represents 63% of the total, followed by whole milk, with 27%, and skimmed milk consumed by 10% of the sample. Meanwhile, based on other specific characteristics, the most consumed type of milk is long-life milk (UHT) with 42% of the sample, followed by lactose-free milk (21%), and fresh high-quality milk (16%) (Table 2).

**Table 2.** Milk purchasing behavior.

| Variables | Description | % | Variables | Description | % |
|---|---|---|---|---|---|
| Purchase frequency | <1 time | 18% | Milk fat percentage | Whole | 27% |
| | 1 time | 33% | | Semi-skimmed | 63% |
| | 2 times | 25% | | Skimmed | 10% |
| | 3 times | 12% | Type of milk | UHT milk | 42% |
| | >3 times | 12% | | Lactose-free milk | 21% |
| Place | Supermarket/Organized large scale distribution | 93% | | High quality milk | 16% |
| | | | | Fresh milk | 10% |
| | Retail | 5% | | Microfiltered milk | 3% |
| | Local producers | 2% | | Organic milk | 3% |
| | | | | Raw milk | 2% |
| | Automatic dispenser | 1% | | Mountain milk | 1% |
| | | | | Special milk | 0% |
| | Local market | 0% | | Flavored milk | 0% |

The respondents who had bought milk were asked to indicate what attributes determine their choice. The three attributes that were most selected by consumers were: the expiry date (18%), the origin (17%), and the price (14%). In order to understand better how the different attributes are significant, consumers were asked to rank, on a five-point Likert scale (from 1 = not important to 5 = very important), the importance of each attribute. The factors that result, by consumers, more important were the expiry date (4.02), the origin (3.80), and the organoleptic characteristics (3.72) (Figure 2).

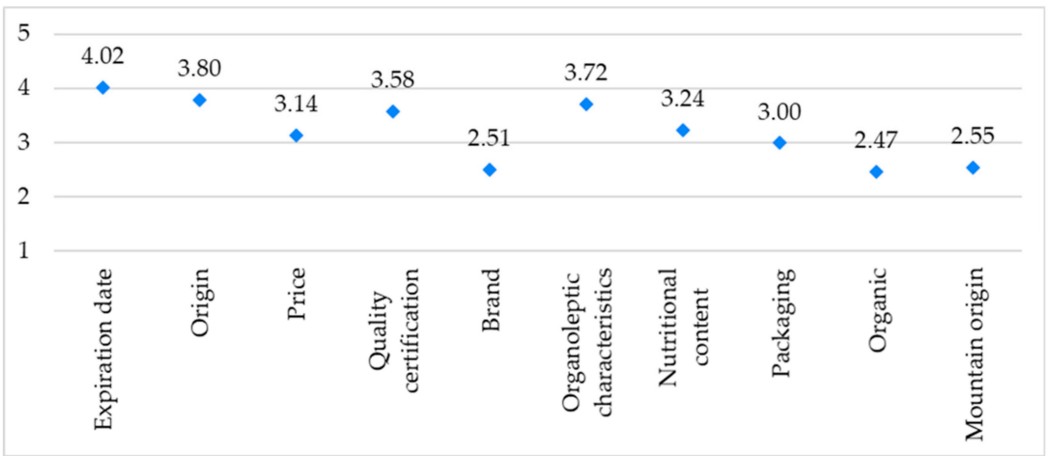

**Figure 2.** Factors influencing the purchase of milk (Likert scale).

When going to examine the answers regarding functional foods, it results that only 34% of the sample is aware of this type of product. Indeed, the concept of functional food has rapidly extended in recent years, but consumers have had little time to gain familiarity with the phenomenon. In addition, this shows a lack of awareness of consumers regarding this type of product, especially due to the absence of unanimously accepted definitions [54–57].

It is important to note that, after being given a definition of functional foods, 79% of the participants have consumed them at least once. This shows a lack of awareness of consumers regarding this type of product.

Those who have never consumed functional foods did not do so because they do not believe they need them (46%), but because they were unaware of their existence (24%) and because they consider them exclusively a fashion (20%). Furthermore, among those who have never consumed them, 80% do not think to purchase this type of product in the future. Meanwhile, those who bought functional

foods did so to fight a specific problem (32%), to prevent a specific problem (30%), and just out of curiosity (28%) (Figure 3).

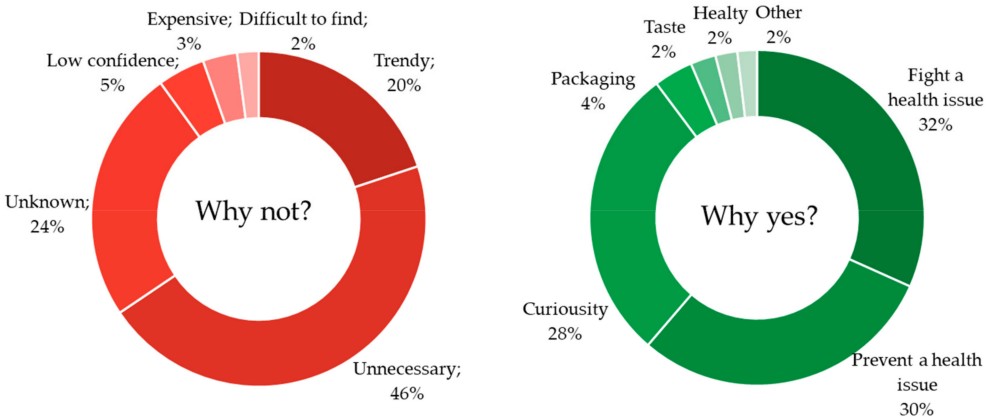

**Figure 3.** Reasons for buying/not buying functional foods.

Among those who purchased functional foods, 81% said they wanted to continue buying these products. Regarding the question—*Have you ever consumed functional milk?* within consumers who bought functional foods, 65% of the sample had consumed functional milk. These data are in line with the positive trend of the functional milk market [58]. Specifically, 10% of the sample tried functional milk only once, 26% rarely consume it, 22% occasionally consume it, 19% consume it frequently, and finally, 22% consume it several times a week. The type of functional milk that is most consumed is highly digestible milk (82%). The spread of this type of milk is linked to the percentage of people with an impaired ability to digest lactose; in fact, according to the National Institutes of Health, approximately 65% of the human population have a reduced ability to digest lactose [59].

Among the remaining types, the main ones are: milk integrated with live or probiotic lactic ferments (5%), milk enriched with vitamins (5%), and milk enriched with omega 3 (4%). In total, 35% of the sample does not consume functional milk mainly because they are not interested in this type of product (80%). The last section of the questionnaire concerned the elicitation of the WTP to purchase A2 milk. Table 3 shows the distribution of the answers for each price offered. Of the sample, 68% declared that they were willing to pay 1.89 EUR for the purchase of A2 milk, also accepting the subsequent proposal of 1.99 EUR; meanwhile, 21% were willing to pay only the first amount. Relating to the part of the sample that refused the initial offer of 1.89 EUR, 4% were willing to pay this product at a lower price of 1.79 EUR, while 6% also refused the second offer.

**Table 3.** Distribution of willingness to pay relative to the price offered.

| Offered Price (Per Liter) | NO-NO | NO-YES | YES-NO | YES-YES | TOTAL |
|:---:|:---:|:---:|:---:|:---:|:---:|
| 1.89 | 6% | 4% | 21% | 68% | 100.0% |

*3.2. The Model Results*

The results of the investigation have so far been presented from a descriptive point of view. The econometric model aimed at analyzing the WTP will be presented below. The estimate of the first model in an enlarged form, that is, in which all the explanatory variables listed above are inserted, has shown that some of these variables are not statistically significant, i.e., other than zero. They were, therefore, eliminated by obtaining a better restricted model based on the information criteria than the previous one, while not significant socioeconomic variables were not excluded from consideration. The final model is represented in Table 4.

**Table 4.** Estimation of the model ranges. Expressed WTP intervals.

|  | Coefficient | Std. Error | z | *p*-Value | |
|---|---|---|---|---|---|
| Const | 0.67680900 | 0.03663100 | 18.48000 | $3.20 \times 10^{-76}$ | *** |
| Age | −0.00136817 | 0.00347544 | −0.39370 | 0.6938 | |
| Female | 0.00030886 | 0.00703401 | 0.04391 | 0.9650 | |
| Education | 0.00433093 | 0.00406125 | 1.06600 | 0.2862 | |
| Employment | 0.00025090 | 0.00285916 | 0.08775 | 0.9301 | |
| Marital | −0.01282760 | 0.00899398 | −1.42600 | 0.1538 | |
| Family | −0.00873134 | 0.00783287 | −1.11500 | 0.2650 | |
| Income | 0.00079954 | 0.00295356 | 0.27070 | 0.7866 | |
| Region | −0.00882986 | 0.00649790 | −1.35900 | 0.1742 | |
| Residence | 0.00214044 | 0.00664031 | 0.32230 | 0.7472 | |
| Place | −0.00615074 | 0.00656307 | −0.93720 | 0.3487 | |
| Milk fat | 0.00182818 | 0.00633099 | 0.28880 | 0.7728 | |
| Milk type | 0.00143654 | 0.00196022 | 0.73280 | 0.4637 | |
| Price | −0.01776970 | 0.00295812 | −6.00700 | $1.89 \times 10^{-9}$ | *** |
| Origin | 0.01117710 | 0.00293769 | 3.80500 | 0.0001 | *** |
| High quality | 0.00828936 | 0.00322040 | 2.57400 | 0.0101 | ** |
| Organic | 0.00929496 | 0.00301003 | 3.08800 | 0.0020 | *** |
| Know functional foods | −0.00268019 | 0.00714359 | −0.37520 | 0.7075 | |
| Buy functional foods | 0.02680540 | 0.00793646 | 3.37700 | 0.0007 | *** |
| Buy functional milk | 0.00252637 | 0.00725002 | 0.34850 | 0.7275 | |

Significance: *** at 1%; ** at 5%.

According to the model, the significant variables that may affect consumers' WTP of respondents for the purchase of the A2 milk are: "Price", "Origin", "High quality", "Organic", and "Buy functional foods". It is possible to obtain a parametric estimate (conditioning) of the value of WTP through the estimation of the model developed; the results are presented in Table 5.

**Table 5.** Descriptive statistics of the variable estimated WTP (in EUR).

| | |
|---|---|
| **Media** | 2.073 |
| **Median** | 2.1323 |
| **Standard Deviation** | 0.14157 |
| **Coefficient of Variation** | 0.068294 |

The results suggest that surveyed consumers are willing to pay a premium of 20 cents in respect to fresh lactose-free milk.

## 4. Discussion

The present findings showed that Italian consumers' intentions to purchase A2 milk were not influenced by sociodemographic variables, but were best explained by the intrinsic (i.e., quality) and extrinsic attributes (i.e., method of production) of the product. These results are in contrast to the study provided by Bimbo et al. [60], which showed how the variables gender and age contribute to explain consumers' acceptance of functional foods.

Starting from the variable "Price", it is significant but with a negative coefficient. This indicates that those who base the choice of which milk to buy on price are willing to pay less for A2 Milk. In line with the literature, this variable is one of the most important extrinsic attributes influencing consumers' purchasing decisions for food products. In accordance with different studies [61,62], the price exerts a negative influence on the decision-making process for individuals. In such circumstances, consumers may also know the health benefits of nutraceutical products, but that does not always translate to purchase behavior.

Consistent with previous studies [63,64], the present model confirmed that a product's origin influences the choice of consumers. In fact, the "Origin" variable is significant and positive.

As Caroprese et al. [65] highlighted, the local origin is perceived as a relevant attribute since the consumer's ethnocentrism induces a preference for food originating from local provenance. This indicates that the people that are careful of the origin of the product are willing to pay more than the others.

Another reason consumers give for their WTP is the presence of organic and quality certification schemes. It appears that consumers' attitudes to both certifications converge towards a perception of high-quality food, for the purchase of which they are willing to pay more. In details, the variable "Organic" indicates that the people who buy this category of milk with a brand that certifies its supply chain are people willing to pay more than the others. This result is in line with the study of Aschemann-Witzel et al. [66], who suggest that organic consumers can be a target group for functional foods characteristics in food as well. Indeed, consumers of functional foods have a similar concept of health and wellbeing to organic consumers [67]. Health is a crucial reason for buying organic food [68–70] in addition to reasons like environmental concerns. Organic foods are usually perceived as healthy as functional foods.

In addition, the results show that those who are attentive to the quality certification scheme will instead be willing to pay more for a product certified and with high-quality characteristics. Indeed, the variable "Quality" is significant and carries the expected sign. The results suggest how food quality could affect consumers' choices in the food market. In agreement with the work of Grunert [71], the quality aspect is consistently important, exerting a positive influence in the decision-making process for individuals, but only when it is perceived as high enough for the consumer to be willing to pay the price demanded in the shop [72]. Clearly, the more information consumers have about quality certification, the more credence they give these systems [73–75].

Finally, the significance and positivity of the variable "Buy functional foods" indicate that those who buy functional foods are willing to pay a higher price. This aspect has also been confirmed by Zou and Hobbs [76] and Carfora et al. [77], which affirms that acceptance of this type of product is closely linked to the trust that the consumer places in them. For this reason, those who have already purchased them in the past are more willing to buy them again than those who have never done it. In addition, Nystrand and Olsen [78], who conducted a quantitative study on a representative sample of Norwegian consumers ($N = 810$) eating functional foods, found that intention was positively associated with consumption frequency, which implies that prior experience with functional foods generates future intention to consume functional foods.

At last, the study also captures the price that consumers are willing to pay for A2 milk. The sample interviewed has indicated an average WTP of about 2.07 EUR per liter. From this result, we can deduce that the premium price for this innovative milk is 20 cents with respect to fresh lactose-free milk. These results are in line with the previous studies which estimate the WTP for functional food products or for organic products. In fact, for this type of product, the interviewed sample has recognized a premium price [79–81]. It is necessary to consider that this average is the expected value of the WTP, given the personal and family characteristics of the person interviewed and the other explanatory variables considered in the estimate; it could be an overestimation of the actual WTP of the sample interviewed due to the problems of the affirmative answer and of the anchoring to the initial value (problems concerning the elicitation method adopted) [82,83].

## 5. Conclusions

Nowadays, in the dairy sector, a hot topic is the growing popularity of A2 beta-casein milk. Farmers in many countries of the world have started to produce A2 milk, especially to meet the growing demand in what is considered to be a healthier alternative to "conventional" or "standard" dairy that contains A1 protein. In fact, the milk protein genetic variation in the original cattle, such as Bruna Alpina and Pezzata Rossa breeds, dual-purpose beef cattle breeds of European origin, was influenced in the past by human movements from different regions as well as by strong crossbreeding. Recently, A2 milk has been rediscovered through the recovery of the original genetic heritage and has been

introduced as a natural functional food. However, there is some debate about the health benefits of A2 milk concerning its classification as a functional food. Despite the uncertainty of the health benefits of this type of milk, it is important to highlight that it is produced by traditional breeds of cattle that are adapted to the marginal areas, such as mountain regions. In other words, the production of milk from these cattle breeds, with original genetic heritage, could be a virtuous model of the sustainable management system (pasture system of mountain areas) to which the consumer lately seems to turn in an increasingly conscious way. In fact, traditional livestock systems may be a source of important services for the human society, including the conservation of livestock biodiversity and of the traditional landscape of meadows and pastures, which is highly valuable from the social, environmental, and economic points of view.

This study highlights that consumers are willing to buy A2 milk, recognizing for it a premium price. The findings obtained from the estimation depend on the characteristics of the respondent and other explanatory variables considered in the estimation. Therefore, we can say that A2 milk sold at that price is a market outlet. With the current milk production situation, the empirical evidence leads to suggest the possibility of strengthening the sector through the diversification of the product. In an extreme conclusion, remember that the production of A2 milk is a valuable opportunity for milk producers, especially breeders who practice extensive farming in mountain areas, but there is a need to implement supply chain integration between the various stakeholders to affect the combination of supply and communication and product promotion. The degree of knowledge and information concerning this food type plays an important role in the choice of purchase, especially considering the role that the production of A2 milk, made from traditional breeds, plays in the support of local economies and the maintenance of biodiversity.

Finally, we should mention the major limitations of this study. Firstly, it is worth emphasizing that this type of milk is not very present on the Italian supermarket shelves. Consequently, a problem commonly present in a large number of WTP measurement surveys is that stated rather than actual preferences are given by participants, resulting in over-ambitious estimations of WTP. Secondly, considering the hypothetical nature of our elicitation method, more research is also needed using non-hypothetical methods to test the robustness of our findings.

Considering the explorative nature of this study, these results should be considered only a springboard for future research in a growing domain.

**Author Contributions:** The authors take part to the research work providing the following contributions: introduction, G.B.; materials and methods, D.B.; investigation, G.S.; data curation, G.S.; descriptive statistics, G.S. and A.F.; model results, D.B. and G.B.; discussion, D.B., G.B., A.F.; Conclusion D.B., A.F., G.B. and G.S.; supervision, A.F.; funding acquisition, A.F. All authors have read and agreed to the published version of the manuscript.

**Funding:** This research was funded by RDP Marche 2014/2020-Submeasure 16.1-Support for the establishment and management of EIP operational groups on agricultural productivity and sustainability.

**Acknowledgments:** The authors wish to thank members of the Operational Group (OP) I-MILKA2—Innovation in dairy production using A2A2 Milk and assessment of the beneficial effects on human health (Project ID n: 29228).

**Conflicts of Interest:** The authors declare no conflict of interest.

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
