# Peer review of "Is There a Promising Market for the A2 Milk? Analysis of Italian Consumer Preferences†"

_sustainability, doi:10.3390/su12176763_

Round 1

Reviewer 1 Report

The text and research are very interesting, but I have a few questions or doubts. They relate to the following lines of text:

  • 175-16: fist bind or first bind?
  • 216-217: You have written "variables connecting to the consumption of milk", but the collected data refer to purchasing behaviour, not to consumption.
  • Figure 2: When we use a five-point scale (from 1 to 5), the OY axis should start at point 1, not 0.
  • 317-318 and 366-368 - I'm not sure how to understand the sentence: "The results suggest that surveyed consumers are willing to pay a premium of 0.20 cent respect conventional product."
    • What did you mean: 0,20 cent = 0,002 Euro? 0,20 Euro = 20 cents? or 20%?
    • What was the reference value - the price of conventional milk in Italy?

Author Response

Answer to the comment from Referee 1

The authors thank the referee for useful comments. The comments and suggestions are valuable and very helpful for revising and improving our manuscript.

The following text includes the answer to the referee’s comments. We have addressed all the points and made most of the changes suggested. Our changes made to the manuscript are in red and highlighted. Please see the revised manuscript and the following answers.

The text and research are very interesting, but I have a few questions or doubts.

They relate to the following lines of text:

1) 175-16: fist bind or first bind?

ANSWER:

Thank you, we have corrected the word “fist” in the revision. We apologize for the oversight.

2) 216-217: You have written "variables connecting to the consumption of milk", but the collected data refer to purchasing behaviour, not to consumption.

ANSWER:

Thank you for this advice. We are grateful for your information. We have corrected this sentence in the revision. The variables considered in our analysis correspond actually to respondent purchasing behaviour on milk.

3) Figure 2: When we use a five-point scale (from 1 to 5), the OY axis should start at point 1, not 0.

ANSWER:

Thank you for your suggestion. You are right, it is better to eliminate the zero in figure 2.

4) 317-318 and 366-368 - I'm not sure how to understand the sentence: "The results suggest that surveyed consumers are willing to pay a premium of 0.20 cent respect conventional product."  What did you mean: 0,20 cent = 0,002 Euro? 0,20 Euro = 20 cents? or 20%?

What was the reference value - the price of conventional milk in Italy?

ANSWER:

Thank you, we have corrected the price in the revision. We apologize for the oversight. We refer to 20 cents. In our analysis, conventional milk refers to lactose-free fresh milk.  We believe we have not described in the paper what we meant by conventional milk. Thanks for the advice. As a consequence, we have specified this aspect in the revised paper. In addition, the price of the conventional milk in Italy derived from a market analysis from similar products, id est different brands of fresh lactose-free milk. The market price was used as a starting point since consumers use reference prices as a threshold to evaluate the selling price of products, which on its turn impacts subsequent purchase decisions. Successively, the design and the ranges of the bid value sets were determined and tested from the focus group discussion and the pre-test survey.

We would like to express my sincere thanks to the anonymous referee who made constructive recommendations and comments on different issues that are now included in the manuscript.

Reviewer 2 Report

In my opinion presented manuscript has no scientific soundness and is limited to surveying consumer nutritonal preferences.

Author Response

Answer to the comment from Referee 2

In my opinion presented manuscript has no scientific soundness and is limited to surveying consumer nutritonal preferences.

ANSWER:

The authors thank the referee for the comments. Please see the following answers.

Considering the attention on A2 milk provided by the scientific community (as shown in Figure 1) we believed that investigating the perception of a consumer on this new product on the market could be challenging. Understanding the demand for this new differentiated product constitutes a cornerstone for further analysis of consumer choices and price competition. Consumer demand in fact drastically changed in recent years: consumers take more interest in aspects such as nutrition, health, and quality of the foods. Consequently, with declines in per capita milk consumption, and changes in consumer preferences, the dairy industry has to be creative and innovative in developing products to increase milk sales. In addition, related to the consumers, the case of Brazilian consumers provided by Medes et al., constitutes one of the few studies on the perception of milk a2.

Our paper wants to fells this gap and contributes to the scientific literature.

Reviewer 3 Report

This paper presents the results of an analysis conducted on over 1,200 questionnaires on the WTP for A2 functional milk. The results show that: (i) variables such as price, product's origin, certification of the supply chain, previous purchase of the functional milk are able to explain the WTP and (ii) consumers are willing to pay a premium of 0,20 cent to buy A2 milk, with respect to the conventional product. 

While the paper is very well written and has several interesting results, I have some concerns and suggestions. 

The first concern is on the validity and internal consistency of the questionnaire. The authors should provide some appropriate tests on this issue. Moreover, it would be useful to provide a description of the representativeness of the sample selected. Some references to support the methods adopted to distribute the questionnaires are useful. 

The second is that it is not clear which is and how is selected the starting price used to compute the price premium for A2 milk. 

The main suggestion is to split the sample according to some individual characteristics of the respondent to see whether the impact of the significant variables on the WTP is heterogeneous and to test their difference. 

Author Response

Answer to the comment from Referee 3

The authors thank the referee for useful comments. The comments and suggestions are valuable and very helpful for revising and improving our manuscript.

The following text includes the answer to the referee’s comments. We have addressed all the points and made most of the changes suggested. Our changes made to the manuscript are in red and highlighted. Please see the revised manuscript and the following answers.

This paper presents the results of an analysis conducted on over 1,200 questionnaires on the WTP for A2 functional milk. The results show that: (i) variables such as price, product's origin, certification of the supply chain, previous purchase of the functional milk are able to explain the WTP and (ii) consumers are willing to pay a premium of 0,20 cent to buy A2 milk, with respect to the conventional product. 

While the paper is very well written and has several interesting results, I have some concerns and suggestions. 

1) The first concern is on the validity and internal consistency of the questionnaire. The authors should provide some appropriate tests on this issue. Moreover, it would be useful to provide a description of the representativeness of the sample selected. Some references to support the methods adopted to distribute the questionnaires are useful. 

ANSWER:

Thank you for your comment and suggestion. We have included more details about these aspects. According to NOAA report (Arrow et al., 1993) the choice of the sample specific design and dimensions are a difficult technical matter this requires an adequate and rational sampling guide. In fact, the questionnaire was pre-tested before the survey was launched. Pre-testing was conducted through a small focus group composed of 25 persons assembled to discuss its understanding of and reaction to the questionnaire prior to the study. Persons including in the pre-test used to consuming both milk and functional food. This is due to the fact that the questionnaire finished with a set of questions designed to obtain the WTP for A2 milk respect to fresh lactose-free milk. We choose this product because today A2 milk is promoted and sold in supermarkets as an alternative for people who struggle to digest common varieties of cows' milk and in some countries, it has been introduced as a functional dairy food due to the alleged natural health benefits. The purpose of the pre-test was to assess if there is a need to make modification on the designed questionnaire. Based on the pre-test, some questions of the survey were restructured and the willingness to pay section was polished so that it became easier to understand by the respondents.

Relating to the second concern, to ensure the representativeness of the sample, the main selection criteria were the consume of milk and consume functional foods. In addition, the sample is representative of the population for age, sex, education and income in Italy. According to Fujita et al. 2005 in order to ensure the statistical reliability of WTP estimations at least 400 samples for a double-bound is needed. In our analysis we have 899 interviewed.

Finally, respondents were sent a unique link to the website on which they simply had to click to start the online survey. This unique link ensured that the same person could not participate more than once.

To conclude we agree with the referee that some references in order to support the methods adopted to distribute the questionnaires are useful. Thus, we have introduced new references as:

  • Alberini, A.; Kanninen, B.; Carson, R.T. Modeling response incentive effects in dichotomous choice contingent valuation data. Land Econ. 1997, 73, 309–324, doi:10.2307/31;
  • Arrow, K., Solow, R., Portney, P. R., Leamer, E. E., Radner, R., & Schuman, H. (1993). Report of the NOAA panel on contingent valuation. Federal register, 58(10), 4601-4614;
  • Brouwer, R.; Langford, I.H.; Bateman, I.J.; Turner, R.K. A meta-analysis of Wetland Contingent valuation studies. In The Stated Preference Approach to Environmental Valuation: Volume III: Applications: Benefit-Cost Analysis and Natural Resource Damage Assessment; 2018; pp. 301–311 ISBN 9781351881579;
  • Dezwaef, J., Cracco, E., Demanet, J., Desmet, T., & Brass, M. (2019). Beyond asking: Exploring the use of automatic price evaluations to implicitly estimate consumers’ willingness-to-pay. PloS one, 14(7), e0219251;
  • Fujita, Y.; Fujii, A.; Furukawa, S.; Ogawa, T. Estimation of Willingness-to-Pay ( WTP ) for Water and Sanitation Services through Contingent Valuation Method ( CVM ) A Case Study in Iquitos City , The Republic of Peru. JBICI Rev. 2005, 10, 59–87;
  • Herriges, J.A.; Shogren, J.F. Starting point bias in dichotomous choice valuation with follow-up questioning. J. Environ. Econ. Manage. 1996, 30, 112–131, doi:10.1006/jeem.1996.0008.
  • Huang, C. H., & Lee, C. H. (2014). Consumer willingness to pay for organic fresh milk in Taiwan. China agricultural economic review;
  • Mauracher, C., Procidano, I., & Valentini, M. (2019). How product attributes and consumer characteristics influence the WTP, resulting in a higher price premium for organic wine. Sustainability, 11(5), 1428;
  • Wright, K. B. (2005). Researching Internet-based populations: Advantages and disadvantages of online survey research, online questionnaire authoring software packages, and web survey services. Journal of computer-mediated communication, 10(3), JCMC1034.

2) The second is that it is not clear which is and how is selected the starting price used to compute the price premium for A2 milk. 

The main suggestion is to split the sample according to some individual characteristics of the respondent to see whether the impact of the significant variables on the WTP is heterogeneous and to test their difference. 

ANSWER:

We agree with the Referee and we thank him/her for pointing this out.  In fact, one of the problems of the CVM includes a hypothetical and starting point for price bidding. The double-bounded dichotomous choice contingent valuation model that we have applied in our paper and pre-testing can reduce the biases. We have introduced some details relating to these aspects in the revised version of the manuscript. The initial bid is derived from a market analysis from similar products, id est different brands of fresh lactose-free milk. The market price was used as a starting point since consumers use reference prices as a threshold to evaluate the selling price of products, which on its turn impacts subsequent purchase decisions. Successively, the design and the ranges of the bid value sets were determined and tested from the focus group discussion and the pre-test survey.

We would like to express my sincere thanks to the anonymous referee who made constructive recommendations and comments on different issues that are now included in the manuscript.

Reviewer 4 Report

The article, “Is there a promising market for the A2 Milk? Analysis 2 of Italian consumer preferences”, by Deborah Bentivoglio, Adele Finco, Giorgia Bucci and Giacomo Staffolani seeks to estimate the premium price that consumers are willing to pay for A2 milk. The authors concluded that consumers are willing to buy the A2 milk recognizing for it a premium price.

The manuscript has been written with a clear flow of headings and informative explanation of each section. The authors considered presenting their research findings with conscientious. They have used a good mix of figures and tables through the whole document.

Below are some suggested amendments and questions for the authors:

  • Line 11- the “policies” has been used only once in the abstract and it has not been addressed anywhere else through the manuscript i.e. introduction. This needs to be defined e.g. what policies had been referred to? Or removed.
  • Authors mentioned that A2 milk has appeared in Italy (Line 98) but not very presented on the Italian supermarket shelves (Line 406). Were all of the asked milk products (UHT milk, fresh milk and etc.) through the survey been categorised under A2 milk (had the label)? My reflection from the presented results is related to the “milk purchasing behaviour” but not the A2 milk product. It is beneficial to have a clarification somewhere within the whole draft, maybe Line 99, that how (special logo?) A2 milk have been presenting in Italy market?
  • Line 64- “benefits of human health” is quit broad it needs to be supported by a reference and at least few cited examples like affect risk of diabetes and/heart disease.
  • Line 72- the presented literature search was not comprehensive enough i.e. only one database has been used. To support the health benefits that A2 milk has or has not I suggest to consider including a published review or a systematic one supporting the presented done literature review (figure 1)
  • I understand study the consumers’ behaviours is a challenging topic! But I like to have further information regarding my question pleas. Authors mentioned the milk consumption has been on the decline (Line 10). I understand this study was a part of a bigger project-I am keen to know whether the amount of consumed milk has been collected at any stage. Only considering “the frequency of buying milk” can be misleading in studying the consumers’ consumption behaviour. In more details, a consumer can buy milk only once a week but in bulk saying four 2litre milk containers, while another consumer can purchase milk three times a week but each time purchase in smaller packages (e.g. 250ml). How this can be addressed?
  • Line 275- Figure 3 can be omitted because there are only two categories (yes, no), which can be easily be presented in words.
  • Line 369- please define bio products

Minor amendments

The manuscript has to be proof read with a careful consideration for checking 1) typos, 2) applying abbreviations and 3) some of the used terms. There are several examples which they have been written differently, or have not been written in full first time used, such as:

  • Line 102, milk a2 and Line105, A2 milk, Line 113 A2-milk, Line 326, A2 Milk
  • Line 362, functional food, Line 364, FF
  • Line 27- EU
  • Line 218- GDO
  • Euros or €
  • Line 369 & 371, willingness to pay or WTP?
  • Line 146, CV, is this different from Line 149, CVM?
  • Line 93- there is no need for presenting HVN as it has been only used once in the manuscript.
  • Line 175- fist bind
  • Line 210- higher and higher income group
  • Line 217-less than one a week- should be said less than once a week
  • Line 37-The numerical superscript has been inserted in an incorrect place- the give description is related to raw milk not the drinking milk.
  • Line 233 and Line 296 used tense is not right.
  • Line 256- table 2, 2 times, 3 times, >3 times
  • Line 64- the sentence begins with “the interest in this type...” is better to go before the first sentence of Line 72, where studies related to A2 milk has been summarized.
  • Line 271- it important to… should have been said it is important to…
  • Line 271- it is suggested to replace the sample with e.g. participants
  • Line 356- indicate that those… it doesn’t need to be written in bold and italic format.
  • Line 351- only one study has been cited [51]- no need to be written in plural

Author Response

Answer to the comment from Referee 4

The authors thank the referee for useful comments. The comments and suggestions are valuable and very helpful for revising and improving our manuscript.

The following text includes the answer to the referee’s comments. We have addressed all the points and made most of the changes suggested. Our changes made to the manuscript are in are in red and highlighted. Please see the revised manuscript and the following answers.

The article, “Is there a promising market for the A2 Milk? Analysis 2 of Italian consumer preferences”, by Deborah Bentivoglio, Adele Finco, Giorgia Bucci and Giacomo Staffolani seeks to estimate the premium price that consumers are willing to pay for A2 milk. The authors concluded that consumers are willing to buy the A2 milk recognizing for it a premium price.

The manuscript has been written with a clear flow of headings and informative explanation of each section. The authors considered presenting their research findings with conscientious. They have used a good mix of figures and tables through the whole document.

Below are some suggested amendments and questions for the authors:

1) Line 11- the “policies” has been used only once in the abstract and it has not been addressed anywhere else through the manuscript i.e. introduction. This needs to be defined e.g. what policies had been referred to? Or removed.

ANSWER:

Thank you for your comment and suggestion. We have removed the word “policies” from the abstract. However, it was referred to as the quota system policy (Line 30-31), the main policy instrument in the EU milk sector.

2) Authors mentioned that A2 milk has appeared in Italy (Line 98) but not very presented on the Italian supermarket shelves (Line 406). Were all of the asked milk products (UHT milk, fresh milk and etc.) through the survey been categorised under A2 milk (had the label)? My reflection from the presented results is related to the “milk purchasing behaviour” but not the A2 milk product. It is beneficial to have a clarification somewhere within the whole draft, maybe Line 99, that how (special logo?) A2 milk have been presenting in Italy market?

ANSWER:

Thank you for your observation. The referee is right to point out that A2 milk has appeared in Italy but it is not very present on the Italian supermarket shelves. To date, in the Italian market, there is already different kinds of milk that have the characteristics of A2 milk. However, they cannot be labelled as "A2 milk" since the a2 Milk Company, founded in New Zealand in 2000, already uses the registered brand a2 Milk™, to identify their products on the market. Due to the lack of a clear logo and the inability to apply this label by Italian producers (Intellectual propriety of a2 Milk Company), the consumer has difficulty identifying A2 milk respect to conventional milk. In addition, due to the controversial debate on the benefit of A2 milk consumption, this product is still not very wide speared on the market.

Consequently, our analysis firstly focused on consumer-buying behaviour for conventional milk, secondly a CVM was applied for A2 milk because of this type of milk is a new product without a market (in fact the contingent valuation is used in order to create a hypothetical market for a non-market good in which the consumer expresses willingness to pay for buying it).

In order to clarify all these issues, we have sought to improve the manuscript (See lines 62-66; Line 103; Lines 108-111).

3) Line 64- “benefits of human health” is quit broad it needs to be supported by a reference and at least few cited examples like affect risk of diabetes and/heart disease.

ANSWER:

Thank you for your comment and suggestion. We have improved the references for the human benefits, by adding the works of:

  • Ho S., et al. “Comparative effects of A1 versus A2 beta-casein on gastrointestinal measures: a blinded randomised crossover pilot study”. European Journal of Clinical Nutrition 68.9 (2014): 994-1000;
  • Jianqin S., et al. “Effects of milk containing only A2 beta casein versus milk containing both A1 and A2 beta casein proteins on gastrointestinal physiology, symptoms of discomfort,and cognitive behavior of people with self-reported intolerance to traditional cows’ milk”. Nutrition Journal 15.1 (2015): 45
  • KamiĹ„ski S., et al. “Polymorphism of bovine beta-casein and its potential effect on human health”. Journal of Applied Genetics 48.3 (2007): 189-198.
  • Kirk, B., Mitchell, J., Jackson, M., Amirabdollahian, F., Alizadehkhaiyat, O., & Clifford, T. (2017). A2 Milk enhances dynamic muscle function following repeated Sprint exercise, a possible ergogenic aid for A1-protein intolerant athletes?. Nutrients, 9(2), 94.
  • Milan, A. M., Shrestha, A., Karlström, H. J., Martinsson, J. A., Nilsson, N. J., Perry, J. K., ... & Cameron-Smith, D. (2020). Comparison of the impact of bovine milk β-casein variants on digestive comfort in females self-reporting dairy intolerance: a randomized controlled trial. The American Journal of Clinical Nutrition, 111(1), 149-160.

4) Line 72- the presented literature search was not comprehensive enough i.e. only one database has been used. To support the health benefits that A2 milk has or has not I suggest to consider including a published review or a systematic one supporting the presented done literature review (figure 1)

ANSWER:

Thank you for your comment and suggestion. We agree with the referee that it would be nice to consider a new database. Thus, We have included in figure 1 also the documents by year from Science Direct database, using the keywords "β-casein" AND "milk. The new figure includes a total of 15.504 documents respect to 2586 documents only from the Scopus database included in the previous version of the paper.

Please see the new Figure 1.

5) I understand study the consumers’ behaviours is a challenging topic! But I like to have further information regarding my question pleas. Authors mentioned the milk consumption has been on the decline (Line 10). I understand this study was a part of a bigger project-I am keen to know whether the amount of consumed milk has been collected at any stage. Only considering “the frequency of buying milk” can be misleading in studying the consumers’ consumption behaviour. In more details, a consumer can buy milk only once a week but in bulk saying four 2litre milk containers, while another consumer can purchase milk three times a week but each time purchase in smaller packages (e.g. 250ml). How this can be addressed?

ANSWER:

Thank you for your comment and suggestion. The decline in milk consumption was revealed by analysis from scientific reports as the "EU agricultural outlook 2018-2030" and "ISMEA report" (See lines 40-43). Furthermore, the quantity of milk purchased was initially taken into account in the analysis but was subsequently discarded in the pre-test phase and it was decided to consider only "the frequency of milk purchase". In fact, most of the testers stated that they mainly consume the 1 litre and 0.5-litre formats and considered the presence of the two questions to be redundant. However, as the reviewer well understood, this research is the first of a series of analyzes aimed at investigating the consumption of milk and functional foods, and in the future, we will follow the advice and we will try to formulate the demand related to consumption in a more complete way.

6) Line 275- Figure 3 can be omitted because there are only two categories (yes, no), which can be easily be presented in words.

ANSWER:

Thank you for your comment and suggestion. We have cut Figure 3.

7) Line 369- please define bio products

 ANSWER:

Thank you for your comment and suggestion. We have corrected the sentence with “organic product”.

Minor amendments

The manuscript has to be proof read with a careful consideration for checking 1) typos, 2) applying abbreviations and 3) some of the used terms. There are several examples which they have been written differently, or have not been written in full first time used, such as:

Line 102, milk a2 and Line105, A2 milk, Line 113 A2-milk, Line 326, A2 Milk

ANSWER:

Corrected. We thank the Referee for pointing this mistake. We have changed with A2 milk

Line 362, functional food, Line 364, FF

ANSWER:

Thank you for your comment and suggestion. We corrected the sentence at line 364 with “functional food”.

Line 27- EU

ANSWER:

In response to this comment we have changed EU with European Union.

Line 218- GDO

ANSWER:

In response to this comment we have changed GDO with organized large scale distribution.

Euros or €

ANSWER:

Thank you for your comment and suggestion. We have modified Euros in €.

Line 369 & 371, willingness to pay or WTP?

ANSWER:

Thank you for your comment and suggestion. We introduced the willingness to pay (WTP) at line 117, and then we used the acronym WTP

Line 146, CV, is this different from Line 149, CVM?

ANSWER:

Thank you for your comment and suggestion.  The model used is the same Contingent Valuation Method (CVM) and we have corrected at line 146 with (CVM).

Line 93- there is no need for presenting HVN as it has been only used once in the manuscript.

ANSWER:

Thank you for your comment and suggestion. We have removed the acronym (HVN)

Line 175- fist bind

ANSWER:

Thank you for your comment and suggestion. We have fixed the error.

Line 210- higher and higher income group

ANSWER:

We have changed the sentence in “Income range: the variable that takes value from 1 to 6 and corresponds to increasingly high-income brackets”. Thank you for your comment.

Line 217-less than one a week- should be said less than once a week

ANSWER:

In response to this comment we have changed the sentence in “less than once a week”

Line 37-The numerical superscript has been inserted in an incorrect place- the give description is related to raw milk not the drinking milk.

ANSWER:

The numerical superscript refers to the drinking milk, which includes Raw milk, whole milk, semi-skimmed and skimmed milk containing no additives.

Line 233 and Line 296 used tense is not right.

ANSWER:

Thank you for your comment and suggestion. We have corrected both the sentences.

Line 256- table 2, 2 times, 3 times, >3 times

ANSWER:

Thank you for your comment and suggestion. We have fixed the error.

Line 64- the sentence begins with “the interest in this type...” is better to go before the first sentence of Line 72, where studies related to A2 milk has been summarized.

ANSWER:

We agree with the reviewer and have moved the sentence at line 71.

Line 271- it important to… should have been said it is important to…

ANSWER:

We agree with the reviewer and have revised the sentence as follows “It is important to”.

Line 271- it is suggested to replace the sample with e.g. participants

ANSWER:

Thank you for your comment and suggestion. We have replaced it with “participants”.

Line 356- indicate that those… it doesn’t need to be written in bold and italic format.

ANSWER:

We agree with the reviewer and have removed the bold and italic format.

Line 351- only one study has been cited [51]- no need to be written in plural

ANSWER:

We agree with the reviewer and have revised the sentence as follows: “In agreement with the work of Grunert”.

We would like to express my sincere thanks to the anonymous referee who made constructive recommendations and comments on different issues that are now included in the manuscript.

Round 2

Reviewer 2 Report

I still remain my opinion about low scientific value of presented manuscript.